# Combination of the Modified Loop Technique and De Vega Annuloplasty in Dogs with Mitral Regurgitation

**DOI:** 10.3390/ani12131653

**Published:** 2022-06-27

**Authors:** Takuma Aoki, Takashi Miyamoto, Naoyuki Fukamachi, Seiya Niimi, Yao Jingya, Yoshito Wakao

**Affiliations:** 1Laboratory of Small Animal Surgery, Department of Veterinary Medicine, School of Veterinary Medicine, Azabu University, Sagamihara 252-5201, Kanagawa, Japan; dv2005@azabu-u.ac.jp (S.N.); dv2103@azabu-u.ac.jp (Y.J.); wakao@azabu-u.ac.jp (Y.W.); 2Department of Cardiology and Respieratory Disease, Azabu University Veterinary Teaching Hospital, Azabu University, Sagamihara 252-5201, Kanagawa, Japan; 3Kodama Kyodo Hospital, Setagaya, Tokyo 156-0052, Japan; yonomiyataka@icloud.com; 4Gunma Chidren’s Medical Center, Shibukawa 377-8577, Gunma, Japan; n.fukamachi7786@gmail.com

**Keywords:** canine, mitral valve, valve disease, cardiac surgery

## Abstract

**Simple Summary:**

Dogs, like humans, may develop valvular disease, which is particularly common in older, small-sized dogs and is one of the most common causes of canine death. Canine valvular disease generally has a good prognosis. However, approximately 30% of dogs develop congestive heart failure, particularly when the mitral valve is affected, and most dogs with severe congestive heart failure die within a year of pulmonary edema. Although cardiac surgery with extracorporeal circulation can be performed in dogs and can significantly prolong survival, the dogs’ small size is challenging from a surgical perspective, and the success rate depends on the surgeon’s skill. Mitral valve repair in dogs involves suturing cardiac tendons using polytetrafluoroethylene sutures, which are slippery and difficult to ligate at the targeted length. Moreover, the appropriate length of the artificial tendon in dogs has not been determined. The mitral annulus surrounding the mitral valve also needs to be stitched down, but the amount that should be stitched down in dogs is not known because prosthetic valve rings were not manufactured for dogs. Due to the lack of reports detailing surgical procedures in dogs, we describe in detail a surgical technique for canine mitral valve repair.

**Abstract:**

Background: Detailed surgical techniques for treating canine mitral regurgitation have not been previously reported. Method: This case series included six consecutive client-owned dogs with mitral regurgitation. All dogs underwent a combined protocol, including the modified loop technique and De Vega annuloplasty (MODEL surgery), in 2021. Artificial loops covering 80% of the length of the strut chordae tendineae were used for chordal replacement. Mitral annuloplasty was subjectively performed, targeting the circumference of the septal leaflet. Results: The breeds were Chihuahua-mixed breed, Spitz, Pomeranian, Cavalier King Charles Spaniel, and Chihuahua, with average ages and weights of 11.4 ± 2.3 years and 5.49 ± 2.98 kg, respectively. The aortic cross-clamp, pumping, and surgery times were 64.0 ± 7.5 min, 168.5 ± 39.1 min, and 321.0 ± 53.1 min, respectively. After MODEL surgery, left atrial-to-aortic ratios significantly decreased from 2.20 ± 0.18 to 1.26 ± 0.22 (*p* < 0.01), and left ventricular end-diastolic internal diameter normalized to body weight significantly decreased from 2.03 ± 0.26 to 1.48 ± 0.20 (*p* < 0.01). In all cases, the clinical signs disappeared or improved significantly. Conclusions: MODEL surgery increased mitral valve coaptation, normalized heart sizes, and significantly improved clinical signs in dogs with mitral regurgitation.

## 1. Introduction

Valvular disease is one of the most important diseases of old age in dogs and humans. Myxomatous mitral valve disease (MMVD) accounts for 75% of all heart diseases in veterinary medicine [1]. MMVD is more common in older, smaller dogs, such as Cavalier King Charles spaniels, Chihuahuas, and Dachshunds [2], and male dogs are 1.5 times more likely to develop the disease than female dogs [1]. Cavalier King Charles Spaniels develop MMVD at a relatively young age, but up to 85% of smaller breeds show valve lesions by 13 years of age [1]. Although canine MMVD has a good prognosis, as in humans, approximately 30% of dogs develop heart failure due to mitral regurgitation (MR) secondary to MMVD [3]. Most patients with heart failure, particularly those with severe disease, die within 1 to 2 years even with aggressive medical treatment [4,5].

Mitral valve repair under extracorporeal circulation using an artificial heart-lung machine has recently been performed in dogs [6,7,8,9]. In humans, this procedure includes chordal replacement using expanded polytetrafluoroethylene (ePTFE) sutures for elongated or ruptured chordae tendineae (CT), ring annuloplasty using a prosthetic valve ring for an enlarged mitral annulus, and valve leaflet resection for an elongated leaflet [10]. Mitral valve replacement is another option, but patients who undergo this procedure require longer thromboprophylaxis, and there is an increased risk of infection or valve dysfunction compared to mitral valve repair. Therefore, in comparisons based on the risk of left ventricular dysfunction, infection, and long-term outcomes, mitral valve repair is the first choice unless the valve lesion is severe [11,12]. The ePTFE suture is an ideal artificial CT; however, it is slippery and challenging to ligate at the targeted length. In addition, the correct length of artificial CT in dogs has not been determined, except in our experimental study [13]. In mitral annuloplasty, a prosthetic valve ring is used in humans; however, to date, no prosthetic valve ring has been manufactured for dogs. Therefore, surgeons perform purse-string suturing (De Vega annuloplasty) in the mitral annulus under visual inspection based on their own experience [7,8]. Valve leaflet resection is an important procedure in humans but is not performed in dogs because the valve leaflets are fragile, and the sutured site might tear. Thus, canine mitral valve repair is performed with chordal replacement using artificial CT and mitral annuloplasty with purse-string sutures [7,9]. However, detailed indications for canine mitral valve repair have not been previously determined, and a subjective assessment is performed by surgeons based on their individual experience. Therefore, surgical outcomes depend on the skills of the surgeon.

We previously developed a modified loop technique that overcomes the slippery nature of ePTFE sutures and numerically expresses the optimal length of artificial CT in experimental dogs [13]. In addition to the modified loop technique, we performed De Vega annuloplasty along the posterior aspect of the mitral annulus up to the right and left fibrous trigones, where the partial prosthetic valve ring was applied in humans [14] to provide posterior remodeling. Uechi et al. showed excellent outcomes with canine mitral annuloplasty by targeting the circumferential diameter of the aortic annulus by preoperative echocardiography [9]. However, in our De Vega annuloplasty, the degree of stitching was intraoperatively determined according to the circumference of the septal leaflet with reference to the Golden Standard in humans [10]. This study aimed to combine the above two methods and evaluate the therapeutic effects of this combined approach in clinical cases.

## 2. Materials and Methods

Consecutive dogs that underwent mitral valve repair for MR due to MMVD at Azabu University Veterinary Teaching Hospital between January 2021 and January 2022 were enrolled in the study. The owners provided written informed consent for surgery. Mitral valve repair was performed by a combination of the modified loop technique and De Vega annuloplasty (hereinafter MODEL surgery) in all cases.

Preoperative and postoperative assessments included measurement of vertebral heart size (VHS) by thoracic radiography and evaluation of the E and A waves, left atrial-to-aortic diameter ratio (LA/Ao), and body weight-normalized end-diastolic left ventricular diameter (LVIDdN) by echocardiography using a 6 or 12 MHz phased-array transducer (Vivid E9; GE Healthcare Co., Ltd., Tokyo, Japan). In addition, the septal leaflet lengths were measured at coaptation (Sc) and end-diastole (Sd) to calculate the coaptation length (CL): CL = Sd−Sc. The CL-index was calculated by dividing CL by the mitral valve short-axis dimension (MVd) at the end-systole. The vertical distance between the annular line and the closing point (Vd) at the end-systole was measured for evaluation of the CT length (Figure 1) [15]. The Vd was measured with the left atrial side set to negative and the left ventricular side set to positive.

For the modified loop technique, 6–12 mm loops were made every 1 mm using CV-6 ePTFE sutures (Gore-tex suture; W. L. Gore & Associates G.K., Tokyo, Japan) and sterilized with ethylene oxide gas before the surgery according to our previous report [13]. Preanesthetic medication included atropine sulfate (0.025 mg/kg, SC; Atropine Sulfate; NIPRO ES PHARMA Co., Ltd., Oosaka, Japan) administered, and fentanyl (5 μg/kg, IV; Fentanyl injection 0.5 mg; Janssen Pharmaceutical K.K., Tokyo, Japan) administered after 5 min of oxygenation. Two minutes after fentanyl injection, midazolam (0.2 mg/kg, IV) was slowly injected, and after induction with propofol (6 mg/kg, IV, to effect; Propofol; Mylan Pharma Co., Ltd., Osaka, Japan), the dogs were intubated. Subsequently, maintenance anesthesia was performed using isoflurane (2%–3%; Isoflurane; Mylan Pharma Co., Ltd., Osaka, Japan). In addition, cefazolin (25 mg/kg, IV; additional doses every 2 h thereafter; Cefamezin; Astellas Pharma Inc., Tokyo, Japan), lansoprazole (1 mg/kg, IV; Takepron; Teva Takeda Pharma Ltd., Nagoya, Japan), dexamethasone (0.1 mg/kg, IV; Dexamethasone injection A; Nippon Zenyaku Kogyo Co., Ltd., Fukushima, Japan), maropitant (1 mg/kg, SC; Cerenia injection; Zoetis JP., Tokyo, Japan), and sivelestat (2 mg/kg/h, CRI; Siverestat NA for injection; Mylan Pharma Co., Ltd., Osaka, Japan) were administered. Fentanyl (15–20 μg/kg/h, CRI) was used for perioperative pain management. In cases showing hypotension with isoflurane inhalation, the anesthesia was converted to a continuous rate infusion of propofol (0.3–0.4 mg/kg/min). Muscle relaxation was performed with rocuronium bromide (0.5 mg/kg, IV; additional 0.1 mg/kg IV every 40 min; Eslax intravenous; MSD K.K., Tokyo, Japan), and mechanical ventilation maintained respiratory care.

During the surgery, we continuously monitored intraarterial and intravenous pressures, heart rate, respiratory rate, end-tidal CO_2_ level, rectal and oesophageal temperatures, arterial blood oxygen saturation, and isoflurane concentration. For measurement of intraarterial blood pressure, catheters were inserted into the dorsalis pedis artery or caudal arteries, and systolic, diastolic, and mean arterial pressures were measured. For the measurement of intravenous pressure, a catheter was inserted into the femoral vein, and central venous pressure was measured. Blood samples obtained through the catheter in the artery or vein were used for complete blood count, activated clotting time measurements, and arterial blood gas measurements. Urine volume was monitored over time with an indwelling urethral catheter.

The dogs were held in the right lateral recumbency position, and after disinfecting the surgical areas, the left external jugular vein and left common carotid artery were surgically exposed for cannula insertion. The heart was approached from the fifth intercostal space on the left side, and heparin (200 IU/kg, IV; heparin sodium injection, NIPRO ES PHARMA Co., Ltd., Oosaka, Japan) was administered at the pleural incision. Three minutes later, ACT was measured, and if this was more than 300–400 s, the patient was connected to an artificial heart–lung machine. The body temperature during extracorporeal circulation was set at 29 °C. After opening the chest, a pericardial sac incision was performed 5 mm below the phrenic nerve, and the edges of the pericardial sac were suspended with 2-0 or 3-0 absorbable sutures (Mono Stinger, BEAR Medic Corporation, Ltd., Tokyo, Japan). A root cannula for antegrade cardioplegia induction was placed in the ascending aorta 5–10 mm distal to the sinotubular junction. The cooled cardioplegic solution (Miotector coronary vascular injection, Kyowa Criticare Co., Ltd., Kanagawa, Japan) of 25 Meq/L of potassium chloride was infused in a dose of 30 mL/kg into the left and right coronary arteries through the root cannula after cross-clamping the ascending aorta distal to the root canula. Thereafter, an additional 15 mL/kg was administered when the heartbeat regained or at intervals of 30 min.

After the left atriotomy, which involved performing an incision laterally on the dorsal side of the left atrium, i.e., 3–5 mm ventral to the left pulmonary vein), we performed visual exploration for elongated or ruptured CT. The valve leaflet was retracted with forceps or hooks to visualize the responsible CT in detail. We measured the length of each strut chordae adjacent to the CT, causing MR in each cranial and caudal papillary muscle (Figure 2). Based on our previous report [13], loop lengths were selected by rounding off 80% of the value of the strut chordae length. Three of these loops of the same length were placed in each papillary muscle (i.e., six loops in total) using double-armed sutures with pledgets and CV-6 ePTFE sutures. Two loops were sutured from each papillary muscle to the septal leaflet, and the remaining loop was attached to the mural leaflet with a simple interrupted suture. The loop closer to the midpoint of the valve cusp at the septal leaflet, and the loop at the mural leaflet received additional simple interrupted sutures as needed. For the loop farther from the midpoint of the septal leaflet, a simple interrupted suture was added to the posterior commissure. After chordal replacement, the leak test was performed by filling the left ventricle with cold saline using a 20 mL catheter syringe. We checked for MR and the size of the septal leaflet with the mitral valve in coaptation status. De Vega annuloplasty was then performed with CV-5 ePTFE sutures. SS size pledgets (Nescosuture; Alfresa Pharma Corporation, Oosaka, Japan) were selected for dogs weighing less than 5 kg, and S size pledgets were used for dogs weighing more than 5 kg. In dogs weighing more than 5 kg, a CV-5 ePTFE suture and an S-size pledget were added to the pledget after stitching with horizontal mattress sutures (Figure 3). The purse-string suture was adjusted until it reached the size of the septal leaflet while repeating the leak test. After completion of all intracardiac manipulations, the leak test was performed again, and if MR appeared to persist, the opposing leaflets were stitched with simple interrupted 5-0 polyvinylidene fluoride sutures (Asflex; Konoseisakusyo Co., Ltd., Tokyo, Japan).

After MODEL surgery, the left atrium was closed with double continuous sutures, and the patient was weaned from the artificial heart–lung machine when the body temperature recovered to 36.5 °C. Protamine (Protamine Sulfate 100 mg for I.V.; Mochida Pharmaceutical Co., Ltd., Tokyo, Japan) was administered in equal doses to heparin to neutralize heparin. One-fifth of the total dose of protamine was administered intravenously as a bolus, and if no hypotension occurred, the remainder was administered as a continuous infusion for 5 to 10 min. The aortic root from which the root cannula was removed was closed with a purse-string suture and a sealant patch coated with human fibrinogen. A sealant patch coated with human fibrinogen (TachoSil Tissue Sealing sheet; CSL Behring K.K., Tokyo, Japan) was also applied to the suture site in the left atrium. In some cases, an additional glue-like human fibrinogen sealant (Beriplast P Combi-Set Tissue adhesion; NIPRO ES PHARMA Co., Ltd., Osaka, Japan) was applied to the sutured sites of the aortic root and left atrium, as needed. Subsequently, a drain was placed, and the chest was closed in a routine manner. If the FiO_2_ was 50% but the SpO_2_ was more than 98%, the patient was moved to an oxygen chamber for management. The patients were maintained in a 30%–40% oxygenated cage for 34–48 h. Blood transfusions were administered postoperatively as needed. Antibiotics were administered intravenously with cefazolin sodium for 3 days after surgery (20 mg/kg, q12hr) and orally with cephalexin (15 mg/kg, q12hr; Cefaclear; Kyoritsu Seiyaku Corporation Co., Ltd., Tokyo, Japan) for the next 4 days. Dalteparin sodium was increased to 150 IU/kg every 12 h after drain removal and continued 1 month after surgery. Clopidogrel (1 mg/kg, SID, PO; Clopidgrel SANIK, Nichi-Iko Pharmaceutical Co., Ltd., Toyama, Japan) was started immediately after the presence of intracardiac thrombus was confirmed by postoperative echocardiography or in the second postoperative month and continued until 3 months after surgery. Thromboprophylaxis was usually stopped 3 months after surgery but was continued further if the D-dimer did not reach the normal range.

Statistical analysis was performed using statistical analysis software (BellCurve for Excel, Social Survey Research Information Co., Ltd., Tokyo, Japan). The values are presented as mean ± standard deviation when the Shapiro-Wilk test showed a normal distribution; otherwise, they were expressed with median (minimum to maximum) values. Comparisons before and after surgery were made using the paired *t*-test or Wilcoxon signed-rank test, and values of *p* < 0.05 were considered significant.

## 3. Results

During the study period, 127 MR cases presented to our hospital and 6 dogs underwent MODEL surgery (Table 1): two Pomeranians and one each of Chihuahua, Chihuahua-mixed breed, Cavalier King Charles Spaniel, and Spitz. The animals included two intact male and four female (1 intact female) dogs. The age of the animals was 11.4 ± 2.3 years, and their body weight was 5.49 ± 2.98 kg. Clinical signs included cough (6/6, three were severe), pulmonary oedema (4/6) and loss of weight (1/6). The cause of severe cough was tracheobronchomalacia due to enlarged heart size in three dogs. In the other three dogs, the owners also reported mild to moderate coughing. The MMVD stages were B2 (2), C (2) and D (2) according to the guidelines of the American College of Veterinary Medicine [1]. Pulmonary hypertension was observed in each dog in stages B2 and D, respectively. Electrocardiography was performed on four dogs, with sinus tachycardia in one dog and sporadic monomorphic ventricular premature contractions in another dog. Two dogs with stage D had azotaemia due to high doses of diuretics. Dogs were treated with pimobendan (6), amlodipine (3), spironolactone (4), furosemide (4), ACE inhibitors (3), or torsemide (3). Chest radiography revealed that four dogs with stage C or D MMVD had pulmonary edema and an enlarged left heart. Three dogs with tracheobronchomalacia had trachea and bronchi that narrowed by more than 50% during expiration. Echocardiography showed moderate-to-severe MR, and left atrium and ventricle enlargement; LA/Ao was 2.18 ± 0.22 and LVIDdN was 2.13 ± 0.17. Two patients with stage B2 underwent MODEL surgery due to a severe cough caused by tracheobronchomalacia secondary to left atrial enlargement (*n* = 1) and severe pulmonary hypertension (right ventricular-right atrial gradient: 99.8 mmHg) secondary to high left atrial pressure (*n* = 1).

The aortic cross-clamp time for the MODEL procedure was 65.3 ± 5.4 min, and the total operative time was 312 ± 48.5 min. Intraoperative visual findings included chordal ruptures in the caudal papillary muscles in four dogs, but only chordal elongation in the two remaining dogs. When performing the modified loop technique, the CT of the responsible lesion was not removed. After mitral valve repair by MODEL surgery, saline solution was placed in the left ventricle to check for regurgitation of the mitral valve, and no regurgitation was observed in five dogs, although slight regurgitation was noted in the remaining dogs.

The postoperative follow-up period was 162 ± 126 days. After MODEL surgery, five dogs showed no cough, and the remaining dog had a cough that was 1/10th of its preoperative level. The preoperative intensity of systolic murmur was Grade 5 in five dogs and Grade 4 in one dog; however, it disappeared postoperatively in five dogs, while the remaining dog still had a Grade 2 murmur. MR disappeared or was markedly reduced (Figure 4). Within 1 week of MODEL surgery, three of the six dogs had thrombi in the posterior wall of the left atrium (2/3) or on the artificial CT (1), although these disappeared over time, and there were no clinical signs of thrombosis.

The heart size was significantly reduced, and the CL-index was significantly improved (Table 2): LA/Ao was 2.21 ± 0.37 preoperatively and 1.29 ± 0.16 postoperatively (*p* < 0.01), and LVIDdN was 2.14 ± 0.18 preoperatively and 1.49 ± 0.21 postoperatively (*p* < 0.01). After MODEL surgery, the E wave decreased from 1.22 ± 0.35 m/s to 0.89 m/s (0.57–1.06 m/s), and the A wave increased from 0.89 ± 0.26 m/s to 1.16 ± 0.18 m/s; however, the differences were not significant (*p* = 0.172 and *p* = 0.159, respectively). In addition, the MVd decreased from 2.07 ± 0.26 cm to 1.26 ± 0.21 cm (*p* < 0.001); Vd decreased from −0.31 ± 0.11 cm to 0.31 ± 0.16 cm (*p* < 0.001). Although the CL-index significantly improved from 0.09 ± 0.07 to 0.19 ± 0.08 (*p* = 0.028), CL was 0.17 ± 0.13 preoperatively and 0.32 ± 0.13 postoperatively, these differences were not significant (*p* = 0.076).

Treatment for the heart was not required in four of the dogs. The azotaemia in two dogs with stage D resolved because diuretics could be withdrawn after MODEL surgery. However, one dog continued to undergo treatment with pimobendan because the postoperative LVIDdN was 1.69, which was at the upper end of the normal range. One dog without a thrombus also showed postoperative collapse due to ventricular tachycardia and was treated with sotalol (1.7 mg/kg, q12hr, PO; Sotakol, Bristol-Myers Squibb K.K., Tokyo, Japan). This dog had not undergone electrocardiography before surgery. Six months after surgery, sotalol was discontinued in one patient; however, recurrent collapse occurred due to ventricular tachycardia, and the patient was still undergoing sotalol treatment. Pulmonary hypertension improved after MODEL surgery in two dogs, decreasing from 99.8 mmHg and 77.1 mmHg to 27.5 mmHg and 38.9 mmHg, respectively.

## 4. Discussion

In the present study, we successfully controlled MR in six dogs with MMVD by performing mitral valve repair according to our protocol. After MODEL surgery, which increased the coaptation of the mitral valve, all six dogs survived with a normalized heart size, and their cough disappeared or improved significantly. Unlike previous techniques based on the individual experience of the surgeon conducting the procedure, MODEL surgery allowed objective chordal replacement by intraoperative measurement of strut chordae length and selecting loops of that length after calculating the correct length. In addition, De Vega annuloplasty could be performed with clear intraoperative goals, such as applying the same circumference to the mitral annulus as that of the septal leaflet. The six dogs included dogs of a variety of stages and weights, but all six showed satisfactory outcomes.

Mitral valve repair in humans includes chordal replacement with artificial CT, valve resection, and annuloplasty using a prosthetic valve ring [10]. In chordal replacement for dogs with MR, ePTFE sutures are commonly used. Although ePTFE is the most suitable suture for artificial CT because of its strength and high biocompatibility [16], its slippery nature makes it difficult to ligate the suture at the targeted length. The loop technique [17], which overcomes these disadvantages and can be performed with a small working space, has become popular in humans, but to our knowledge, no report other than our previously published research has described its application to experimental dogs [13]. In addition, our report is the first to mention the appropriate length of artificial CT in dogs with MR [13]. In our six consecutive cases, both septal and mural leaflets were replaced by a loop of 80%, as long as the strut chordae. The use of both leaflets of the same height increased valve coaptation, and the intraoperative leak test showed minimal or no MR in these six cases. In the present study, valve leaflets that had prolapsed into the left atrium were restored into the left ventricle on echocardiography. Annuloplasty with prosthetic valve material is performed to further improve coaptation and long-term durability for mitral valve repair [18,19]. Excessive shrinkage of the annulus may result in mitral stenosis and systolic anterior motion (SAM) in the elongated septal leaflet [20]. In this study, in all six dogs, the mitral annulus was sewn down to the area of the septal leaflet, but no cases developed postoperative SAM or mitral stenosis. In cases with a significantly enlarged annulus, total ring stitching around the whole circumference of the annulus is recommended [19], although the total ring restricts physiologic movement of the mitral annulus and is particularly prone to SAM [14,20]. However, since sufficient coaptation is achieved by only chordal replacement in these case series, partial annuloplasty excluding the septal leaflet was performed. Partial annuloplasty has been associated with a low complication rate [14], including SAM [14,20].

MODEL surgery significantly increased the CL-index, which was reportedly more important than the CL in regulating MR since decreased MVd further improved the coaptation of the mitral valve [15]. However, long-term follow-up is needed for mitral stenosis because stitching may shrink below the physiological size of the annulus using our method. In the present study, the mitral annulus shrunk to 60% of its preoperative size in the cephalocaudal direction. In fact, although E and A waves are proportional to left atrial pressure [21], no significant postoperative reduction was observed in the present study. Although the above should be clarified in future research, the aortic cross-clamp time of our MODEL surgery is shorter than that reported previously and is performed by skilled surgeons in dogs of 60–95 min [9], which may mean that the procedure is simpler.

The risk factor for postoperative thrombus is surgical invasion of the mitral valve and endocardium [22], but it can also be partly attributed to the artificial nature of ePTFE [23]. In addition, postoperative left ventricular inflow is reduced because of the normalized volume of the left atrium, and the reduced blood flow can lead to thrombus formation [24]. In MODEL surgery, the presence of multiple knots on the loops may influence thrombus formation. It was not clear whether it was on the loop knot; however, in one dog, a thrombus was suspected of having formed on the loop in the present study. After this case series, low-dose clopidogrel (0.5 mg/kg, q24hr, PO) was used early in combination with dalteparin sodium unless the patient showed melena or another form of bleeding after drain removal; then, no intracardiac thrombus formation was observed.

This study included one case of ventricular tachycardia with collapse requiring sotalol after MODEL surgery. In dogs with MR, atrial fibrillation increases in proportion to the left atrial diameter and body weight [25]. Ventricular tachycardia is also present, particularly after the onset of congestive heart failure [26]. Mitral annular disjunction, which is characterized by dislodgement between the valve ring and myocardium, is a risk factor for ventricular arrhythmias in humans and is more common in mitral valve prolapse and severe MR cases [27]. Ventricular dilation and stretching with increased myocardial wall strain, the cause of ventricular arrhythmia [28], were resolved postoperatively, but ventricular tachycardia recurred after sotalol withdrawal. Although our patients showed no mitral annular disjunction, chordal traction caused by redundant leaflets and mechanical stimulation of the endocardium are other possible mechanisms underlying ventricular arrhythmias [29]. Mechanical stimulation from chordal replacement or annuloplasty may be the cause of ventricular arrhythmias. Although ventricular tachycardia could have been present before surgery, electrocardiography was not performed in this case. In MR cases with cardiac enlargement, arrhythmias should be assessed before surgery.

Coughing in dogs with MR has been reported to increase proportionally to left atrial enlargement [2]. Although compression of the trachea and bronchi by an enlarged left atrium is thought to cause cough, Singh et al. reported no association between airway collapse and left atrial size in dogs with chronic cough [30]. However, after MODEL surgery, cough disappeared in five of the six dogs in our study. Because chronic tracheobronchitis is also associated with coughing in small, older dogs [31], the presence of chronic bronchitis was suspected in dogs with residual coughing.

The limitations of our study include the small number of cases and the short follow-up period. Moreover, we did not perform the procedure with artificial loop lengths other than 80%. In addition, the length of the artificial CT determined in the empty non-beating heart may not always match the required length in the beating heart [32]. We might also study preoperative echocardiographic determination of the length of artificial CT. Furthermore, in all cases, three loops were placed in each papillary muscle. In other words, only six loops were used in total; however, more loops may be required depending on the condition of the mitral valve. In addition, the area of the septal leaflet was used as a reference for annuloplasty, but this is a relatively subjective method. The patient with post-operative ventricular tachycardia should have been assessed for arrhythmias before surgery. Finally, using our modified loop technique, thrombus formation on the loop was observed in only one case. There were no thrombus-related clinical signs in the dog mentioned above or in the two dogs with left intra-atrial thrombus; however, early aggressive thromboprophylaxis seems to be a necessary requirement in our protocol.

## 5. Conclusions

In conclusion, we combined the modified loop technique with De Vega annuloplasty to perform mitral valve repair, which increased coaptation of the mitral valve and allowed control of MR in dogs with MMVD. We plan to repeat the application of this MODEL surgery in more clinical cases and conduct further long-term follow-up assessments.

## Figures and Tables

**Figure 1 animals-12-01653-f001:**
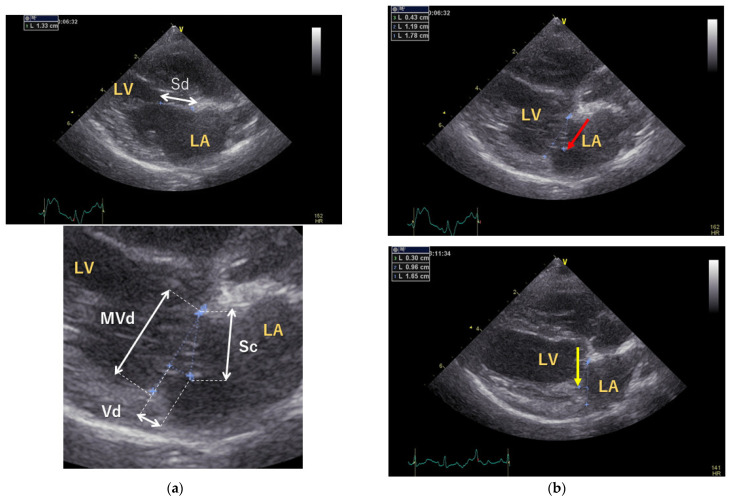
Echocardiographic assessment of mitral valve morphology. (**a**) The septal leaflet lengths were measured at end-diastole (Sd) and coaptation (Sc) using a right parasternal four-chamber view. The coaptation length (CL) was calculated using the following formula: CL = Sd-Sc. The CL-index was calculated by dividing the CL by the mitral valve short-axis dimension (MVd) at the end-systole. The vertical distance between the annular line and the closing point (Vd) at the end-systole. The Vd was measured with the left atrial side set to negative and the left ventricular side set to positive. (**b**) The upper and lower figures show the mitral coaptation of the end-systolic mitral valve before and after surgery. After surgery, the mitral valve height (red arrow) was reduced, and mitral valve coaptation was seen in the left ventricle (yellow arrows). LA: left atrium; LV: left ventricle; Sd: septal leaflet length at end-diastole; Sc: septal leaflet length at coaptation; CL: coaptation length; MVd: mitral valve short-axis dimension at end-systole; Vd: vertical distance between annular line and closing point at end-systole.

**Figure 2 animals-12-01653-f002:**
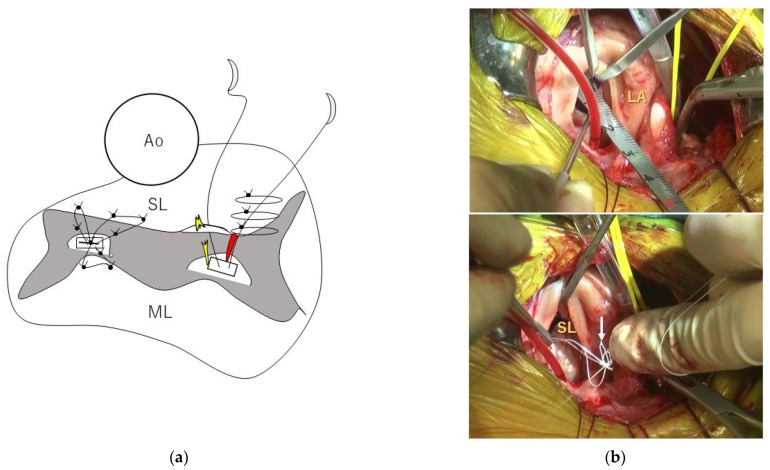
Modified loop technique. (**a**) Schema of the modified loop technique, in which loops of various lengths are pre-made, sterilized, and used during surgery. Further, 80% of the loops of the strut chordae (red) length were selected to repair ruptured or elongated chordae tendineae (yellow). (**b**) Using a ruler, the length of the strut chordae (top) was measured. Then, three loops (white arrow), each 80% of the length of the strut chordae, were fixed to the cranial and caudal papillary muscles with a double-armed suture. Of the three loops, two are fixed to the septal leaflet and one to the mural leaflet with simple interrupted sutures (bottom). The loops of the mural leaflet and the center of the septal leaflet are ligated at two points at the valve leaflet. Ao: Aorta; SL: Septal leaflet; ML: mural leaflet; LA: left atrium.

**Figure 3 animals-12-01653-f003:**
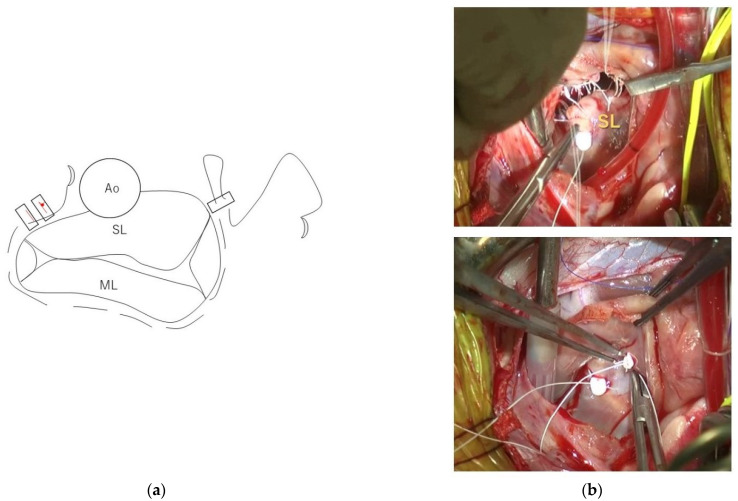
De Vega annuloplasty. (**a**) Schema of De Vega annuloplasty, in which the annulus is sutured down to the size of the septal leaflet. The mitral annulus, except the base of the septal leaflet, is stitched using CV-5 ePTFE sutures with a purse-string suture through pledgets. If the weight was more than 5 kg, a horizontal mattress suture was added to secure it more rigidly. (**b**) Intraoperative photos showing the septal leaflet (top). The mitral valve is visualized at the status of mitral valve coaptation by injecting cooled saline solution into the left ventricle to confirm the size of the septal leaflet or the presence of MR (bottom). SL: Septal leaflet; ML: mural leaflet.

**Figure 4 animals-12-01653-f004:**
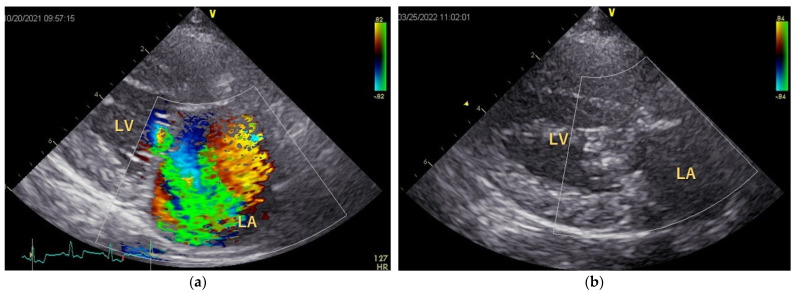
Color Doppler echocardiography. (**a**) Preoperative parasternal four-chamber view showing large turbulence mosaic patterns from the left ventricle to the left atrium, indicating the presence of severe MR. (**b**) Postoperative parasternal four-chamber view shows the disappearance of MR and reduction in the left atrial size. LA: left atrium; LV: left ventricle.

**Table 1 animals-12-01653-t001:** Signalment, ACVIM stage, and clinical signs of 6 cases.

Case	Breeds	Age (Years)	Sex	BW (kg)	ACVIM Stage	Clinical Signs
1	Spitz	12.3	F	6.34	C	pulmonary oedema
2	Chihuahua-mixed breed	14.3	F	3.46	B2	pulmonary hypertension, tracheobronchomalacia, loss of weight
3	Pomeranian	10.4	F	5.3	D	pulmonary oedema, azotaemia
4	C.K.C.S.	7.3	M	10.98	C	pulmonary oedema
5	Chihuahua	12.4	F	2.78	B2	tracheobronchomalacia
6	Pomeranian	11.7	M	4.14	D	pulmonary oedema/ hypertension, tracheobronchomalacia, azotaemia

ACVIM: American College of Veterinary Internal Medicine; C.K.C.S.: Cavalier King Charles Spaniel.

**Table 2 animals-12-01653-t002:** Comparison of radiographic and echocardiographic measurements before and after surgery.

	Pre	Post	*p*-Value
VHS (*n* = 5)	11.5 ± 0.6	10.5 ± 0.6	0.042
LA/Ao (*n* = 6)	2.21 ± 0.37	1.29 ± 0.16	<0.01
LVIDdN (*n* = 6)	2.14 ± 0.18	1.49 ± 0.04	<0.01
E wave (*n* = 6)	1.22 ± 0.35 m/s	0.89 m/s (0.57–1.06 m/s)	0.172
A wave (*n* = 6)	0.89 ± 0.26 m/s	1.16 ± 0.18 m/s	0.159
MVd (*n* = 6)	2.07 ± 0.26	1.26 ± 0.21	<0.001
Vd (*n* = 6)	−0.31 ± 0.12	0.31 ±0.16	<0.001
CL (*n* = 6)	0.17 ±0.13	0.32 ± 0.13	0.076
CL-index (*n* = 6)	0.09 ± 0.07	0.19 ± 0.08	0.028

VHS: vertebral heart size; LA/Ao: left atrial-to-aortic root diameter ratio; LVIDdN: Left ventricular end-diastolic internal diameter normalized to body weight; MVd: mitral valve short-axis dimension at end-systole; Vd: vertical distance between annular line and closing point at end-systole; CL: coaptation length.

## Data Availability

Not applicable.

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
