# Peer review of "Combination of the Modified Loop Technique and De Vega Annuloplasty in Dogs with Mitral Regurgitation"

_animals, 2022, doi:10.3390/ani12131653_

Round 1

Reviewer 1 Report

Dear Authors,

I found your paper very interesting, although below I suggest some corrections:

  • line 78: it seems the reference is incorrectly cited - please check the reference
  • line 214: the animals' age shown in years will be more clear
  • line 215: it is not clear whether all dogs shown all the symptoms or there was variety between the animals; if the latter is more accurate, the symptoms should be presented in a table together with animals' age and MMVD stage according to ACVIM and treatment
  • it is not clear wether the animals had an ECG performed in the diagnostic process? as there was arrhythmia present post-operatively, it is not clear if it appeared after the surgery or was present earlier
  • table 1 should be moved in the text where it is mentioned
  • references 2 and 4 are older than 40 years; although MMVD is a topic of research for many years, I think it is possible to find references more up-to-date; moreover the reference 2 is from 1965 - basing on the development of diagnostic and scientific methods, more accurate papers should be cited

Reviewer 2 Report

Review for animals-1729550

This is an interesting paper concerning the mitral valve repair surgical protocol for mitral regurgitation due to myxomatous mitral valve disease. While the manuscript provides potentially important information and insight into surgical treatment for mitral regurgitation dogs, a few concerns are identified as “major and minor comments to the author”.

Major comments:

1. What is “single” surgical protocol? Does it refer to the loop technique?

You did perform mitral valve repair with two combination technique (modified loop technique and De Vega annuloplasty).

2. I feel that the purpose of this study and the conclusion are different.

Your purpose is to establish a simple and reproducible surgical technique, right?

However, your conclusions, like any other paper so far, are just the course of clinical outcomes and cardiac remodeling indicators.

In order to achieve the purpose of this paper, it may be necessary to evaluate the shortening of the operation time, the postoperative suitability of the chordae tendineae, and the postoperative coaptation of the mitral valve.

3. About the length of chordae tendineae, you judge it by the measurement result at the time of cardiac arrest. Is there any difference from the physiological length at the time of postoperative heartbeat?

I think it is good to evaluate the postoperative chordae tendineae length and the postoperative valve coaptation etc by echocardiography.

4. As you mentioned in the limitations, the annuloplasty does not seem to be a simple and objective method. Also, does the "size" of the anterior leaflet mean the area of the anterior leaflet?

5. I'm sorry for lack of study, but is the notation MODEL surgery common? If it is a unique notation, I think it needs a certain degree of novelty.

6. It is said that there are 6 consecutive cases, but how many cases of MMVD visited the hospital, and 6 of them were indicated for surgery?

7. Regarding the indications for surgery, B2 is also included, but is it an aggressive indication for surgery?

Minor comments:

Title: I think the term "simple" needs to be reconsidered.

Line 22: I think the expression "completely repair" is an overstatement.

Line 27: Is the notation “MODEL” unique?

Line 76: Is the Reference 16 your study?

Line 142: How many chordae tendineae do you make?

Line 202: Was the intracardiac thrombus judged by echo after surgery?

Line 226: Did you remove the original chordae tendineae that were torn and stretched? Or did you keep it as it is?

Line 231: I think this is the part where the results of the surgical procedure are written, but did you find a blood clot during the surgery?

Line 248: “VT” You should not be abbreviated.

Line 265: What do you mean “consistent”?

Line 266: Why are you putting the discussion of ”valve resection” here?

Line 279: You judge it by the measurement result at the time of cardiac arrest. Is there any difference from the physiological length at the time of postoperative heartbeat?

Line 280: Did you asses the postoperative valve coaptation?

Line 286: Please add the reference paper.

Line 293: If you have, please add the A wave value.

Line 300: This looks like a serious limitation. Was the blood clot in the chordae tendineae immediately after the operation (line 231) part of this knot?

Reviewer 3 Report

The authors have carried out a high-level and innovative work for veterinary medicine. Ho alcune richieste di chiarimento.

  • Line 132: can the authors better describe the incision of the pericardial sac and the relationships with the phrenic nerve?
  • Line 133: ..."a root cannula was placed at the aortic root...": can the authors better describe this anatomical detail and the procedure reported? refer to the aortic root, actually located in the ventricular-aortic junction or the first tract of the ascending aorta? can the authors make a synthetic anatomical and surgical consideration towards the coronary branches?
  • Line 138: "After the left atriotomy...": the left atrium has a complex morphology, can the authors describe this passage better? in which precise point has happened the atriotomy?
  • Line 138: "...we performed visual exploration for elongated or ruptured 138 chordae." from the atrial side it is possible to have a limited view of the tendon cords. You can see the broken strings that fit close to the free margin of the flap, but not all the others. Considering the complex anatomy of the tendon cords, can the authors deepen this important passage considering the particular and unique view from the atrial side?

Round 2

Reviewer 2 Report

This is an interesting paper concerning the mitral valve repair surgical protocol for mitral regurgitation in dogs. Since most of the corrections have been made, I hope that the following minor points can be cleared.

Line 176: Is the number of chordae tendineae 6 in all cases, regardless of the condition of the valve?

And are they six of the same length?

Line 376: What is [Author1]?
